

# Effect of thermodenuding on the structure of nascent flame soot aggregates

Janarjan Bhandari[1], Swarup China[1,*], Timothy Onasch[2,3], Lindsay Wolff[3], Andrew Lambe[2,3], Paul Davidovits[3], Eben Cross[2], Adam Ahern[4], Jason Olfert[5], Manvendra Dubey[6], Claudio Mazzoleni[1]

[1]Department of Physics and Atmospheric Sciences Program, Michigan Technological University, Houghton, MI, 49931, USA
[2]Aerodyne Research Inc., Billerica, MA, 01821, USA
[3]Chemistry Department, Boston College, Chestnut Hill, MA, 02467, USA
10 [4]Centre for Atmospheric Particle Studies, Carnegie Mellon University, Pittsburgh, PA, 15232, USA
[5]Department of Mechanical Engineering, University of Alberta, Edmonton, Alberta, T6G 2G8, Canada
[6]Earth and Environmental Sciences Division, Los Alamos National Laboratory, Los Alamos, NM, 87545, USA
[*] Now at: Pacific Northwest National Laboratory, Richland, WA, 99354, USA

15 **Abstract.** The optical properties (light scattering and absorption) of soot particles depend on soot size and index of refraction, but also on the soot complex morphology and the internal mixing with other material at the single particle level. For example, freshly emitted (nascent) soot particles can interact with other materials in the atmosphere, materials that can condense on soot and coat it. This coating can affect the soot optical properties by refracting light, or by changing the soot aggregate structure. A common approach to studying the effect of coating on soot optical properties is to measure absorption 20 and scattering values in ambient air and then measure them again after removing the coating using a thermodenuder. In this approach, it is assumed that: 1) Most of the coating material is removed; 2) charred organic coating does not add to the refractory carbon; 3) oxidation of soot is negligible; and 4) the pre-existing core soot structure is left unaltered despite potential oxidation of the core at elevated temperature. In this study, we investigate the validity of the last assumption, by studying the effect of thermodenuding on the structure of nascent soot. To this end, we analyze the morphological properties 25 of laboratory generated nascent soot, before and after thermodenuding. Our investigation shows that there is only minor restructuring of nascent soot by thermodenuding.

## 1 Introduction

Soot particles are mostly composed of refractory carbonaceous material that forms from incomplete combustion during burning activities (Haynes and Wagner, 1981). A nascent soot particle appears as a fractal-like (sometimes referred to as a 30 lacy) aggregate of small spherules (called nanospheres or monomers) (Buseck et al., 2014) and its structure is scale invariant (Sorensen, 2001). Here, the term nascent is used to refer to freshly emitted soot particles that have minimal or no coating on the nanospheres. The diameter of these nanospheres varies in a range from 10 nm to more than 50 nm depending on the fuel source and combustion conditions (e.g., Adachi and Buseck, 2008; Bambha et al., 2013; China et al., 2014; Park et al.,



2004). During atmospheric processing, soot particles interact with organic and inorganic materials (in the form of aerosol or condensable vapors). During these interactions, soot undergoes morphological changes including for example compaction, coagulation and coating (e.g., China et al., 2015b). Coating or internal mixing in general, changes the optical properties of soot even when the structure of the refractory components remain the same. These changes consequently affect the radiative

forcing of soot (Adachi and Buseck, 2013; Ghazi and Olfert, 2013; Jacobson, 2001; Lack and Cappa, 2010; Liu et al., 2015a; Schnitzler et al., 2014; Van Poppel et al., 2005; Zhang et al., 2008). Several studies have shown that coating of soot by partially-absorbing or non-absorbing materials increases the absorption and scattering cross sections (Cappa et al., 2012; Fuller et al., 1999; Khalizov et al., 2009; Lack and Cappa, 2010; Liu et al., 2015b). These increases are termed as absorption and scattering enhancements ($E_{abs}$ and $E_{sca}$). The enhancement is typically calculated as the ratio of the light absorption or

scattering coefficient of the coated soot to the light absorption or scattering coefficient of the soot after the coating material has been removed (Lack and Cappa, 2010).

Thermodenuders (TDs) that remove the coating by evaporation are often used in the field and the laboratory to study and quantify these enhancements (Bambha et al., 2013; Ghazi and Olfert, 2013; Lack et al., 2012; Xue et al., 2009). During the

thermodenuding process, coated soot particles are passed through a heated column, typically at ~200-300 $^0$C to evaporate volatile coating material, while leaving behind the refractory soot (Huffman et al., 2008; Wehner et al., 2002). The temperature gradient within the TD can result in particle losses due to thermophoretic forces, though these losses can be measured (Huffman et al., 2008). To correctly assess the $E_{abs}$ and $E_{sca}$ using a TD, one needs to make the following assumptions: 1) Most of the coating materials is removed from the soot by the TD; 2) organic coating material does not

transform into refractory carbon due to charring; 3) refractory carbon is not oxidized to a substantial extent under the elevated temperature; and 4) the structure of the refractory soot particle is unaffected, meaning that the thermodenuding process alone does not induce the restructuring of the lacy soot aggregate. In fact, when uncoated soot undergoes structural changes (e.g., compaction of its lacy structure), absorption and scattering change as well, even in the absence of a change of the overall particle mass or composition (China et al., 2015a; China et al., 2015b; Liu et al., 2008; Radney et al., 2014).


Thermodenuding does not remove refractory particulate material, such as some inorganic salts from soot particle and may not remove all of the non-refractory material (Cappa et al., 2013). For example, Liu et al. (2015b) observed that denuded soot still contained heavily coated soot particles although in a smaller fraction, suggesting that the TD does not remove the coating completely, especially for ambient samples. Healy et al. (2015) found that on average only 74% of the mass of

coating material was removed from soot samples after thermodenuding. The mass removal efficiency by TD was even less ( approximately 60%) for wildfire emission samples. Swanson and Kittelson (2010) have also cautioned about semi-volatile particle artifacts due to incomplete removal of evaporated compounds in the TD. Similarly, Knox et al. (2009) found that there was no significant difference in the mass absorption cross section between the themodenuded and non-thermodenuded aged-soot sample compared to fresh soot due to the incomplete removal of coating materials from aged soot particles. For



these reasons, some have argued that the use of thermodenuder may cause biases in measurement by reducing the true value of the optical enhancements (Khalizov et al., 2013).

The elevated temperatures during thermodenuding may also cause charring of some organic matter into refractory, elemental carbon and /or some oxidation of the carbonaceous matter. Charring of organic particulate material into elemental carbon is a known phenomenon under elevated temperature in OCEC techniques (Chow et al., 2004; Countess, 1990). Issues that influence charring include temperatures and residence times, as well as chemical composition. Charring is likely more of an issue for oxidized organics, such as biomass burning and/or SOA, than reduced organics, such as efficient combustion products (i.e., diesel and laboratory flame soot) (Cheng et al., 2011; Khan et al., 2012). Two significant differences between OCEC and thermodenuding include: (1) OCEC techniques typically operates at higher temperatures than TDs, and (2) OCEC charring occurs in a helium atmosphere, whereas thermodenuding occurs in air (i.e. oxidizing environment). Thus, while low temperature (<300 $^0$C) thermodenuding will be less likely to char due to temperature, they may be more susceptible to oxidation due to an oxidizing environment. Oxidation of refractory carbon soot typically occurs at significantly elevated temperatures, but can occur at lower temperatures, especially if catalyzed due to impurities in the soot (Stanmore et al., 2001). Soot oxidation is likely limited in thermodenuding due to the low temperatures and relatively short residence time.

In this study, however, our focus is not to investigate the validity of these first three assumptions, but instead to study the potential effect of thermodenuding on the soot structure (assumption 4). The impacts on the core structure of first coated and then thermodenuded soot have been studied previously for different coating materials (Cross et al., 2010; Ghazi and Olfert, 2013; Xue et al., 2009). For example, Cross et al. (2010) studied the effect of coating and denuding on the soot optical properties using sulfuric acid and dioctyl sebacate (DOS) as coating materials. The study showed that there was a stronger restructuring of the soot core using sulfuric acid compared to DOS. Xue et al. (2009) found significant restructuring of soot particles when the particles were first coated with glutaric acid and then denuded. Ghazi and Olfert (2013) reported the dependence of soot restructuring on the mass amount of different coating material types. These observations show that coating of soot particles and their subsequent thermodenuding can cause a restructuring of soot aggregates. Two potential explanations for the soot restructuring detected during these studies can be: 1) Soot might be compacted during condensation of the coating materials due to surface tension effects (Huang et al., 1994; Tritscher et al., 2011; Zhang et al., 2008). 2) The removal of the coating material may cause compaction during the evaporation process, still due to surface tension effects (Ebert et al., 2002; Ma et al., 2013). We propose a third potential pathway for soot restructuring solely by thermodenuding: Heating during thermodenuding may weaken the bonds between the monomers in the aggregate causing the soot structure to change even when no or minimal coating is present. Specifically this third issue is the subject of our study. With this goal, we analyze the effect of thermodenuding on the structure of laboratory generated nascent soot particles produced from two different fuel sources (ethylene flame and methane flame) and size selected at different mobility diameters. This assessment



is important for evaluating the potential biases that might be introduced by thermodenuding while, for example, estimating the absorption or scattering enhancements of laboratory or ambient soot particles.

## 2 Experiment

### 2.1 Experimental setup and sample collection

We analyzed five pairs of mobility-selected soot samples collected during two different experiments: the Boston College Experiment 2 (BC2) and the Boston College Experiment 4 (BC4). The sampling schematics are shown in the Figure 1a and b. None of the samples were coated with additional external coating material and the minimal coating present on the nascent soot was solely due to fuel residuals accumulated during the combustion and dilution processes.

Three soot sample sets were collected during BC2 from the combustion of ethylene and oxygen using a premixed flat flame burner (Cross et al., 2010). The fuel equivalence ratio (∅) for all the three sample sets was 2.1. A TD (Huffman et al., 2008) was used to remove volatile components from the nascent soot particles. The heating section of the TD was set at 250 $^0$C to vaporize the non-refractory soot components which were absorbed by a charcoal section maintained at room temperature. Particles for a range of mobility diameters ($d_m$) were selected to investigate the effect of thermodenuding on particle size.

Since the soot particles were only minimally coated, the thermodenuded particles had only slightly smaller mobility diameter than the initial particles. For our investigation, we selected three sets of nascent vs. nascent-denuded soot particles with $d_m =$ 153 nm, 181 nm and 250 nm for nascent soot particles and $d_m =$ 151 nm, 175 nm and 241 nm for the corresponding denuded soot particles. Soot particles were collected on 13 mm diameter Nuclepore polycarbonate filters having a pore size of 0.3 $\mu$m (Whatman Inc, Chicago, Illinois, USA). Additional details of the BC2 experimental set-up are provided elsewhere (Cross et

al., 2010).

In addition, we selected two sets of soot samples generated during BC4 from the combustion of methane in an inverted diffusion flame burner (methane and $O_2$ mixture) at $d_m =$ 253 nm and 252 nm for nascent soot particles and $d_m =$ 253 nm and 251 nm for the corresponding denuded soot particles. The ∅ for both sample sets was about 0.7. As in BC2, a Huffman TD

(heating section set at 270 $^0$C) was used to remove the volatile components. For both experiments, the sample flow rate through the TD was 2 LPM resulting in a residence time of 5s in the heating section and 4s in the denuder section. During BC4, unlike during BC2, particles were first mobility size selected by a Differential Mobility Analyzer (DMA, TSI Inc.) and mass selected by a Centrifugal Particle Mass Analyzer (CPMA, Cambustion Ltd.) before thermodenuding. The first set of samples consisted of nascent and nascent-denuded soot, while the second set consisted of nascent-oxidized and nascent-

oxidized-denuded soot. Soot was oxidized by exposure to ozone and hydroxyl radicals in a Potential Aerosol Mass (PAM) oxidation flow reactor (Lambe et al., 2011) and the nascent-oxidized soot was thermo-denuded at a temperature of 270 $^0$C. The set of nascent-oxidized soot samples before and after thermodenuding was included here to investigate the possible difference in the effect of thermodenuding between nascent and nascent-oxidized soot. During BC4, soot particles were



collected on 13 mm diameter Nuclepore filters having a pore size of 0.1 µm diameter (Whatman Inc, Chicago, Illinois, USA).

All the filters were coated with 1.8 nm thick layer of Pt/Pd alloy in a sputter coater (Hummer® 6.2) and imaged with a Hitachi S-4700 field emission scanning electron microscope (FE-SEM). From the FE-SEM images, several morphological

parameters were evaluated using the image processing software ImageJ (China et al., 2014; Schneider et al., 2012).

**2.2 Soot morphological parameters**

Soot particles are aggregates of monomers that exhibit scale-invariant fractal structures (Forrest and Witten Jr, 1979; Sorensen et al., 1992). Soot aggregates can therefore be characterized by a fractal dimension $(D_f)$ in which the mass of the

aggregate $M$ (proportional to the number of monomers $N$ in the aggregate) scales with the ratio of the radius of gyration $(R_g)$ to the radius of the monomers $(R_p)$, as in $M$ (or $N$) $\propto (R_g/R_p)^{D_f}$ (Klein and Meakin, 1989). To quantify the soot morphology, $D_f$ is a commonly used parameter. Lacy soot particles have low $D_f$ values, while compact soot particles have higher $D_f$ values. The $D_f$ of an ensemble of soot particles can be calculated by plotting $N$ vs. $R_g$ (or a surrogate for it). $N$ scales with $R_g$ as a power law with exponent $D_f$ (Köylü et al., 1995b):

$$N = k_g \left(\frac{R_g}{R_p}\right)^{D_f} \tag{1}$$

where $k_g$ is a pre-factor whose value depends on the overlap between the monomers in the aggregate. The relation formulated by Köylü et al. (1995a) was used to estimate $D_f$ with geometric mean diameter $(\sqrt{LW})$ as a surrogate for $2R_g$ :

$$N = k_{LW} \left(\frac{\sqrt{LW}}{2R_p}\right)^{D_f} \tag{2}$$

where $L$ is the maximum length and $W$ is the maximum width (orthogonal to $L$), $K_{LW}$ is a prefactor and $R_p$ is calculated from

the mean of the projected area of the monomer. In general, it is difficult to measure $N$ using an SEM image alone, because only two-dimensional (2-D) projections of the soot particles are typically available. Therefore, $N$ is often estimated from the projected area of the soot aggregate $A_p$ and the mean projected area of the monomers $A_m$ using the relation provided by Oh and Sorensen (1997):

$$N = k_a \left(\frac{A_p}{A_m}\right)^{\alpha} \tag{3}$$

where $\alpha$ and $k_a$ are constants that depend on the overlap between monomers in the 2-D projected image of the particle. In our case, we used $K_a = 1.15$ and $\alpha = 1.09$ for all our nascent and nascent-denuded soot aggregates (Köylü et al., 1995b). This selection of $K_a$ and $\alpha$ values is reasonable since we only studied nascent soot particles that are minimally coated.

In addition to $D_f$, several other 2-D morphological parameters were calculated from the FE-SEM images to investigate

potential changes due to thermodenuding. The calculated parameters included roundness, convexity, aspect ratio $(AR)$ and area equivalent diameter $(D_{Aeq})$. Figure 2a shows the definition of some of these parameters. $D_{Aeq}$ is the diameter of a spherical particle with a projected area equivalent to the projected area of the aggregate. Roundness is calculated from the



ratio of the projected area of the aggregate to the area of the circle having a diameter equal to the maximum projected length $L$ and fully inscribing the projected image of the aggregate (Fig. 2b). Convexity (sometimes termed solidity) is the ratio of the projected area of the particle to the area of the smallest convex hull polygon, in which the 2-D projection of the aggregate is inscribed (Fig. 2c). $AR$ is calculated as the ratio of $L$ to $W$. Higher values of roundness and convexity or lower AR often

correspond to more compact soot particles. In total, 1224 images of individual soot particles were analyzed.

## 3 Results and discussion

We analyzed images from four sets of nascent and nascent-denuded soot sample pairs of different sizes and a fifth set of nascent-oxidized denuded soot. Examples of images of soot particles before and after thermodenuding are shown in Fig. 3.

$N1, N2, N3$ and $N4$ are 4 differently sized nascent soot particles and $D1, D2, D3$ and $D4$ are the corresponding nascent-denuded sets. $N5$-$D5$ is a pair of nascent-oxidized soot before and after thermodenuding. Table1 summarizes the features of the analyzed soot particles. Sets $N1$-$D1$, $N2$-$D2$ and $N3$-$D3$ are the three sets from BC2 while sets $N4$-$D4$ and $N5$-$D5$ are from BC4.

In Table 1, for fuel type, E = ethylene and M = methane. $N$ is the average number of monomers per aggregate estimated in each sample using Eq. (3). $K_g$ values have been estimated using the relation $K_g = K_{L \cdot W} (1.17)^{Df}$ where $\sqrt{LW}/2R_g = 1.17$ has been taken from (Köylü et al., 1995b) and the values of $K_{L \cdot W}$ and $D_f$ have been calculated from a log-log plot using Eq. (2). $d_m$ is the mean mobility diameter (in nm) and $M_{CPMA}$ represents the mean mass of the particle (in fg) as measured by the CPMA. For $D_f$, the term in parenthesis is the standard error calculated from the power fit using Eq. (2).


The largest decrease in the mean value of $d_p$ (by 5.6 %) after thermodenuding is found for the $N3$-$D3$ set. The decrease in $d_p$ could be due to the removal of material volatile at the TD temperature and present on the nascent soot. Also, the mean value of $d_p$ is smaller for the inverted diffusion flame when particles are sampled at the same mobility diameter of 250 nm. This suggests that there was less volatile material present in nascent soot generated from the inverted flame. This could be due to

the different type of fuel as well as Ø. For the inverted diffusion flame of methane, Ø was set at a lower value of ~ 0.7 compared to 2.1 for the pre-mixed flat flame of ethylene. For soot generated by an inverted diffusion flame, Ghazi and Olfert (2013), found no measurable amount of volatile material when the mass was measured by a CPMA before and after thermodenuding while for the nascent soot containing 0.1 mass fraction of non-refractory material (at Ø =2.1) from an ethylene flat flame, Slowik et al. (2007) found that thermodenuding removed only about 0.05 mass fraction of volatile

material.

To investigate whether the soot aggregate restructured after thermal denuding, we first analyze the changes in $D_f$ as summarized in Fig. 4. For all the five sample sets, $D_f$ lies between 1.65 and 1.86 (Table 1). These values of $D_f$ are in





agreement with observations made in previous studies on nascent soot particles produced from different fuel sources e.g., (Dhaubhadel et al., 2006; Sorensen, 2001). Also for all nascent vs. denuded pairs (except for the nascent-oxidized pair: *N5-D5*), there is no significant change (within $1\sigma$) in $D_f$ after thermodenuding (Fig. 4). For the *N5-D5* pair, $D_f$ changes by about 9% (from 1.65 to 1.80) whereas for all other cases, the change is less than 2.3 %. The CPMA data for BC4 sample shows that the mass decreased from 2.37 to 2.34 fg for nascent soot, while for the nascent-oxidized soot of the same mobility size, the mass decreased from 2.41 to 2.18 fg after thermodenuding. The larger decrease in mass for the nascent oxidized soot suggests that coating material on the oxidized soot was removed during thermodenuding. A possible explanation for the increase of $D_f$ after thermodenuding the oxidized soot might be that the soot structure was slightly modified during the evaporation of the coating material. Interestingly, for the BC2 soot samples, there is no significant change in $D_f$ despite the remarkable change in mass (~ 25%) of soot after thermodenuding (see CPMA data in Table 1). This result suggests that for BC2 sample sets, the removal of coating present on nascent soot does not affect the structure of soot. This is most probably due to the chemical composition of organics that were removed by TD. Cross et al. (2010) observed only minor restructuring of soot when dioctyl sebacate coating was removed by thermodenuding, suggesting that the removal of aliphatic compound may have little impact on the restructuring of soot. For soot from a flat flame burner, Slowik et al. (2004) found that the OC content (mass fraction of 0.1) was composed of comparable amount of aliphatic and aromatic compounds at lower ∅ (∅= 1.85) but at higher ∅ (∅>4), the OC content (mass fraction of 0.55) had only minor fraction of aliphatic compounds. We thus hypothesize that the nascent organics on the soot from the BC2 experiments considered here consisted in large fraction of aliphatic compounds.

To further investigate possible morphological changes after thermodenuding, we studied the convexity and roundness of soot particles for all five sample sets. The maximum change in the mean value of roundness occurs for set *N3-D3* (about 18%) and followed by the set *N4-D4* (about 13%). For the other sets, the mean value of roundness changes by less than 10%. For the case of convexity, the maximum change in the mean value occurs for set *N4-D4* (about 8%). For all other sets, the mean value of convexity changes by less than 5%.

In Fig. 5a and 5b we show box and whisker plots for the convexity and the roundness, respectively of the soot particles before and after thermodenuding. The convexity ranges from 0.37 to 0.91, while the roundness ranges from 0.09 to 0.75 (see Table 1 for details). No substantial changes in roundness or convexity are evident after thermodenuding.

In Fig. 6, we show probability distributions of convexity and roundness for all nascent and denuded soot pairs. The solid and the dashed lines represent the mean values for nascent and denuded soot respectively, while the shaded color bands in blue and orange represent one standard deviation. These means and uncertainty bands are calculated with a bootstrap approach, resampling with replacement from the raw data and constructing 100,000 frequency distributions (Wilks, 2011).





For four of the five sets, we observe minor changes in convexity and roundness. For the *N3-D3* pair, the distribution of convexity and roundness peaks at slightly lower values after thermodenuding. The convexity of particles decreases slightly with increasing value of the mobility diameter for both nascent and denuded particles. This suggests that smaller soot particles are more compact compared to larger particles, in agreement with previous studies (Chakrabarty et al., 2006;

Virtanen et al., 2004). Figure 6a also suggests that for smaller mobility diameters, the convexity of soot from the ethylene diffusion flame might be less affected by thermodenuding compared to the larger sized particles. With the methane diffusion flame (*N4-D4* and *N5-D5* sets), particles showed a negligible change in convexity after thermodenuding for both nascent and nascent-oxidized soot (Fig. 6a   *N4-D4* and *N5-D5* respectively).

For completeness, we also investigated the changes in *AR* and $D_{Aeq}$, both show only small changes after thermodenuding (Table 1). Our observations on the 5 sets of soot pairs show that the morphology of nascent soot does not change significantly after thermodenuding. Exceptions are the roundness for *N3-D3* and *N4-D4* sets and the convexity for *N4-D4* pairs, which showed some statistically significant, but still small changes. Roundness and convexity on nascent-oxidized soot pair (*N5-D5*) showed no significant change after thermodenuding. However, the change in $D_f$ (1.65±0.05 to 1.80 ±0.05)

is significant. Since we have only a single set for the nascent-oxidized soot, we are unable to draw with confidence a conclusion on the effect of thermodenuding on such particles. However, the changes in $D_f$ suggest that the different nanophysical properties of the nascent-oxidized soot might indeed result in a higher sensitivity to thermodenuding. In a study on the fragmentation and bond strength of diesel soot, Rothenbacher et al. (2008) made a comparison between soot treated with and without a TD as a function of impact velocity and  found no substantial change in the size distribution and structure

of soot aggregates due to the  thermodenuding. A low-pressure impactor was used to impart velocities up to 300m/s to the soot particles. The TD used in their study had a residence time of 0.43 s and the sample was heated to 280 $^0$C. In another study by Raj et al. (2014), soot fragmentation was observed after thermodenuding in the temperature range of 400-900 $^0$C on both  diesel soot and commercial soot (Printex-U). However, in the lower temperature range, below 500 $^0$C, they found a minor effect on soot fragmentation. Bambha et al. (2013) noticed little effect of thermodenuding at 410 $^0$C (transit time of

~34 s) on the morphology of soot during the removal of oleic acid coating. In another study, Slowik et al. (2007) did not observe any measurable change in the structure of soot when uncoated soot (generated at ∅ = 2.1 and 3.5) was thermodenuded at 200 $^0$C. Our results of the negligible or minor restructuring of thermodenuded soot particles are in agreement with these previous studies suggesting that these results are robust and reproducible. The novel contribution of our study is the selective use of nascent soot from ethylene and methane fuels.

**4 Summary**

In this study, we used scanning electron microscopy to investigate the structure of nascent soot aggregates prior and after thermodenuding in a low-temperature regime (< 270 $^0$C). Our observations based on soot samples from two fuel sources showed only minor restructuring of nascent soot after thermodenuding, irrespective of the fuels. Despite mass losses up to





~29% in the nascent soot after thermodenuding, only minor effects on the structure of nascent soot were detected. No significant change in $D_f$ was observed; the only exception was the fractal dimension of nascent-oxidized soot, although roundness and convexity showed only minor changes also in this case. Since there are only minor changes in the structure of nascent soot, the absorption and scattering cross sections of nascent soot should not change appreciably after

thermodenuding. Future work should focus on the effect of higher thermodenuding temperatures to investigate temperature effect on the structure of nascent soot.

*The image analysis data used in this paper are publicly available on the Digital Commons repository of Michigan Technological University and can be found here: http://digitalcommons.mtu.edu/physics-fp/80*

**Acknowledgements**

This work was supported in part by the Office of Science (BER), Department of Energy (Atmospheric System Research) Grant no. DE-SC0011935 and no. DE-SC0010019, and Atmospheric Chemistry program of the National Science Foundation Grant no. AGS-1536939 to Boston College, 1537446 to Aerodyne Research Inc. S. China was partially supported by a

NASA Earth and Space Science Graduate Fellowships no. NNX12AN97H.

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

**Table 1.** Summary of physical and morphological parameters for the soot particles analyzed.

| Experiment | BC2 | | | | | | BC4 | | | |
|---|---|---|---|---|---|---|---|---|---|---|
| Sample | N1 | D1 | N2 | D2 | N3 | D3 | N4 | D4 | N5 | D5 |
| Fuel type | E | E | E | E | E | E | M | M | M | M |
| #Particles analyzed | 108 | 151 | 113 | 163 | 114 | 109 | 113 | 105 | 122 | 125 |





| $N$ | | 41 | 55 | 121 | 104 | 110 | 153 | 158 | 188 | 155 | 166 |
|---|---|---|---|---|---|---|---|---|---|---|---|
| $d_m$ (nm) | | 153 | 151 | 181 | 175 | 250 | 241 | 253 | 253 | 252 | 251 |
| $M_{CPMA}$ (fg) | Mean | 1.02 | 0.78 | 1.52 | 1.08 | 2.85 | 2.20 | 2.37 | 2.34 | 2.41 | 2.18 |
| | S.D. | 0.03 | 0.03 | 0.05 | 0.04 | 0.14 | 0.13 | 0.11 | 0.13 | 0.11 | 0.11 |
| $D_f$ | Mean | 1.86 | 1.84 | 1.73 | 1.72 | 1.78 | 1.79 | 1.80 | 1.76 | 1.65 | 1.80 |
| | S.E. | (0.05) | (0.04) | (0.05) | (0.06) | (0.08) | (0.05) | (0.05) | (0.06) | (0.05) | (0.05) |
| $K_g$ | Mean | 1.78 | 1.98 | 2.50 | 2.50 | 2.22 | 2.00 | 2.10 | 2.56 | 2.87 | 2.16 |
| | S.E. | (0.04) | (0.03) | (0.05) | (0.05) | (0.08) | (0.06) | (0.06) | (0.07) | (0.06) | (0.06) |
| $d_p$ (nm) | Mean | 33.5 | 31.8 | 26.8 | 25.7 | 32.1 | 30.3 | 23.5 | 22.8 | 23.9 | 23.1 |
| | Median | 33.5 | 32.4 | 26.5 | 25.9 | 32.1 | 28.9 | 23.2 | 22.5 | 23.7 | 23.0 |
| | S.D. | (2.1) | (3.3) | (2.7) | (2.6) | (2.1) | (6.9) | (3.1) | (2.2) | (2.5) | (3.4) |
| | S.E. | (0.21) | (0.27) | (0.26) | (0.21) | (0.20) | (0.66) | (0.30) | (0.22) | (0.23) | (0.31) |
| Roundness | Mean | 0.41 | 0.43 | 0.36 | 0.34 | 0.38 | 0.31 | 0.31 | 0.35 | 0.33 | 0.33 |
| | Median | 0.42 | 0.42 | 0.35 | 0.35 | 0.35 | 0.30 | 0.30 | 0.34 | 0.32 | 0.31 |
| | S.D. | (0.12) | (0.12) | (0.10) | (0.10) | (0.12) | (0.09) | (0.11) | (0.13) | (0.11) | (0.11) |
| | S.E. | (0.01) | (0.01) | (0.01) | (0.01) | (0.01) | (0.01) | (0.01) | (0.01) | (0.01) | (0.01) |
| Convexity | Mean | 0.72 | 0.75 | 0.66 | 0.66 | 0.62 | 0.59 | 0.61 | 0.66 | 0.61 | 0.63 |
| | Median | 0.73 | 0.74 | 0.66 | 0.65 | 0.62 | 0.58 | 0.61 | 0.66 | 0.61 | 0.62 |
| | S.D. | (0.09) | (0.09) | (0.09) | (0.10) | (0.09) | (0.10) | (0.10) | (0.11) | (0.12) | (0.11) |
| | S.E. | (0.01) | (0.01) | (0.01) | (0.01) | (0.01) | (0.01) | (0.01) | (0.01) | (0.01) | (0.01) |
| $D_{Aeq}$ (nm) | Mean | 169 | 181 | 220 | 196 | 255 | 262 | 215 | 230 | 219 | 214 |
| | Median | 171 | 175 | 208 | 189 | 262 | 260 | 199 | 220 | 213 | 202 |
| | S.D. | (33) | (35) | (55) | (40) | (46) | (49) | (54) | (55) | (50) | (59) |
| | S.E. | (3) | (3) | (5) | (3) | (4) | (5) | (5) | (5) | (5) | (5) |



| AR | | | | | | | | | | | |
|---|---|---|---|---|---|---|---|---|---|---|---|
| | Mean | 1.79 | 1.73 | 1.84 | 1.92 | 1.78 | 1.85 | 1.99 | 1.95 | 1.85 | 1.88 |
| | Median | 1.66 | 1.62 | 1.70 | 1.78 | 1.68 | 1.72 | 1.85 | 1.82 | 1.80 | 1.83 |
| | S.D. | (0.51) | (0.42) | (0.49) | (0.51) | (0.57) | (0.50) | (0.60) | (0.60) | (0.50) | (0.50) |
| | S.E. | (0.05) | (0.03) | (0.05) | (0.04) | (0.05) | (0.05) | (0.06) | (0.06) | (0.05) | (0.04) |

*(a)*

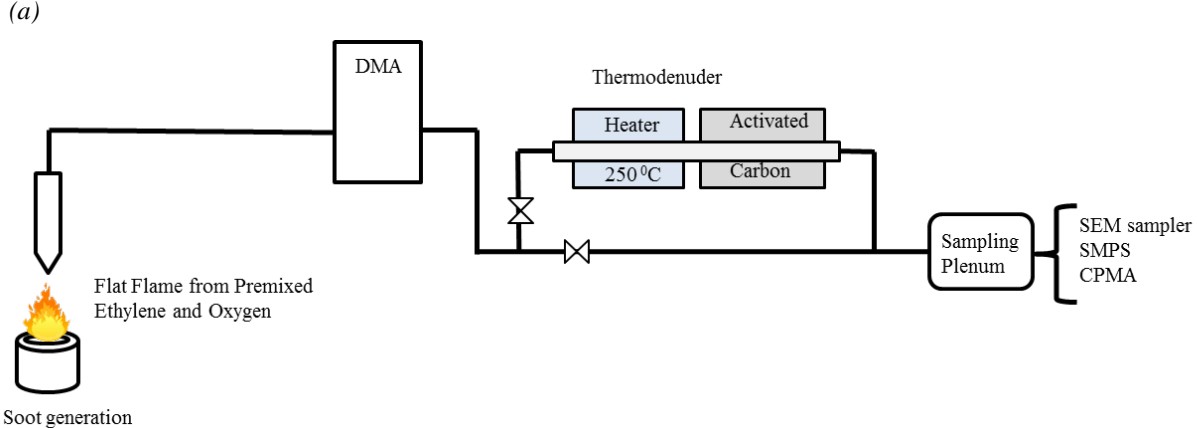

*(b)*

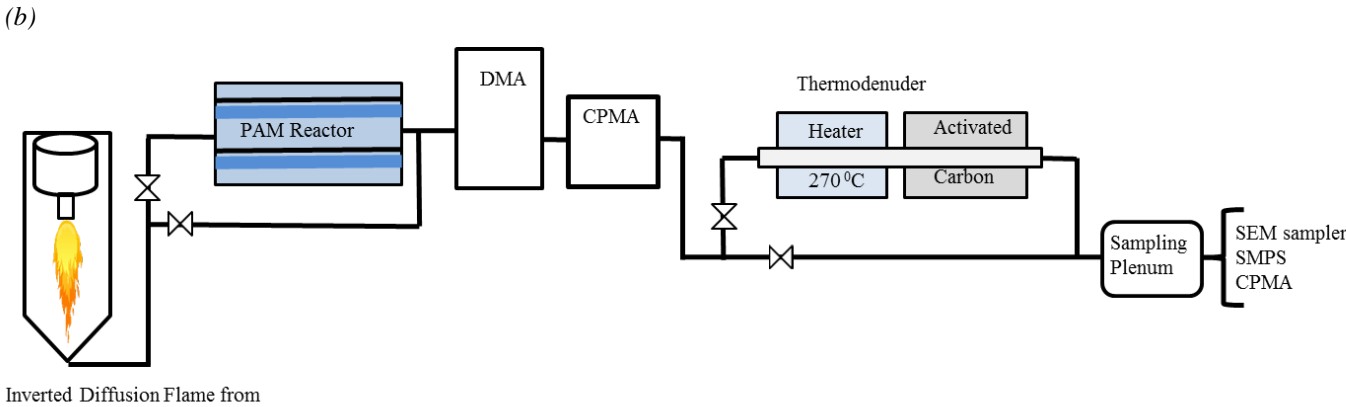

**Figure. 1 (a)** Soot generation and sampling in BC2. **(b)** Soot generation and sampling in BC4.



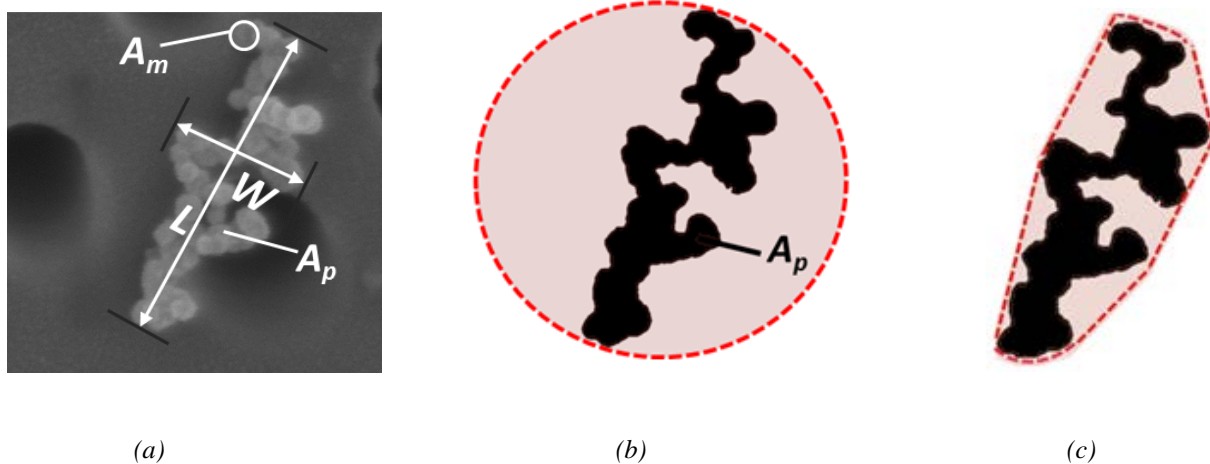

*(a)*            *(b)*            *(c)*

**Figure 2. (a)** Example of SEM image of a soot particle showing the definition of several parameters measured from the binary projected image: maximum projected length $L$, maximum projected width $W$, projected area of monomer $A_m$ and projected area of particle $A_p$. **(b)** Schematic representation of the roundness calculation for the same soot particle shown in (a). **(c)** Schematic representation of the convexity calculation for the same soot particle shown in (a). The pink shades in (b) and (c) represent the equivalent area for circle and convex hull, respectively, for the binary image of soot particle shown in (a).

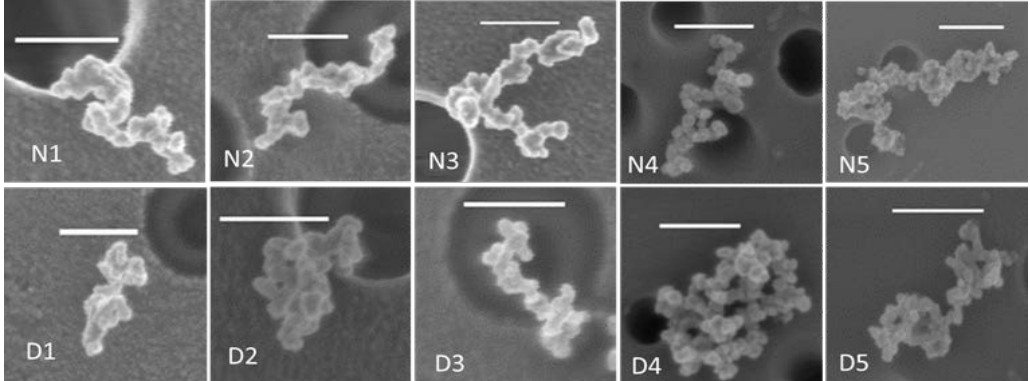

**Figure 3.** SEM micrographs of nascent (N) and thermodenuded (D) soot particles. The white horizontal bar in each micrograph represents a length scale of 200 nm.





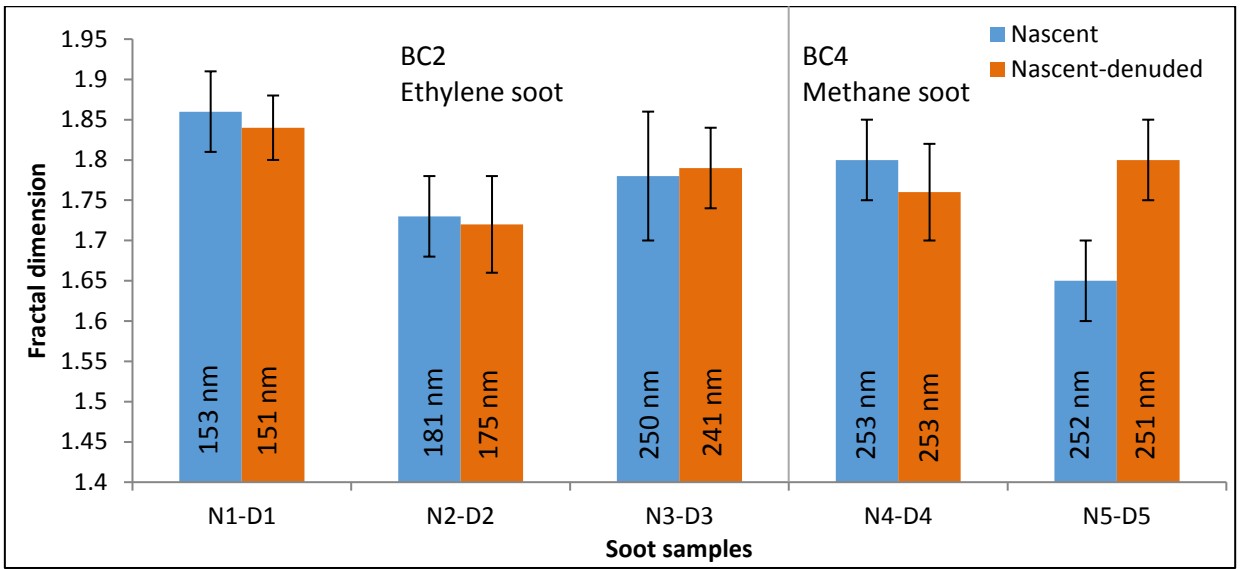

**Figure 4.** Fractal dimension of nascent (in blue) and nascent-denuded (in orange) soot pairs of different mobility sizes. The error bars represent the standard errors.

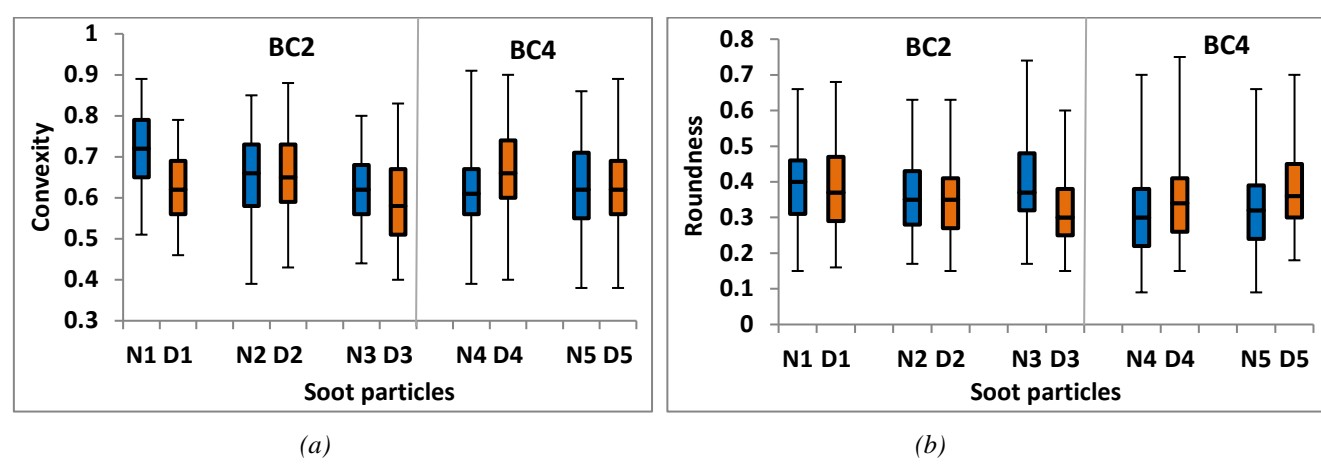

*(a)*          *(b)*

**Figure 5.** Box and whisker plots of **a)** convexity and **b)** roundness. Blue boxes represent the nascent soot and orange boxes represent the nascent-denuded soot. The horizontal bar inside the box represents the median value while the lower part and upper part of the box separated by the horizontal bar represent the first and third quartiles respectively. The lower and upper extremities of the whiskers represent the minimum and maximum values, respectively.





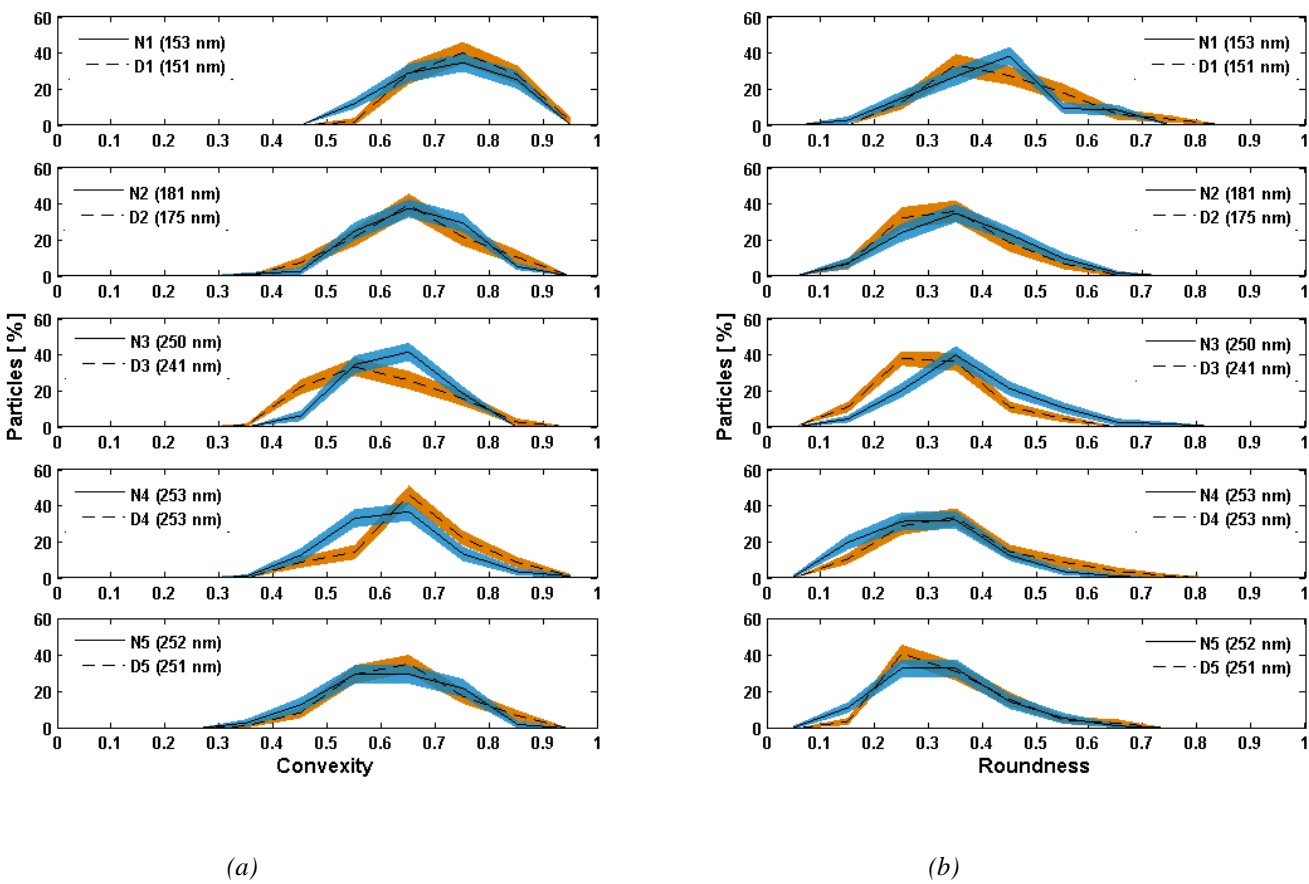

*(a)*                                    *(b)*

5 **Figure 6.** Distributions of **a)** convexity and **b)** roundness for nascent and nascent-denuded soot particles of different sizes
(the mobility diameter is reported in parenthesis in the legends).