# Peer review of "Effect of thermodenuding on the structure of nascent flame soot aggregates"

_Atmospheric Measurement Techniques, 2016_

## Referee Comment (RC1) · Anonymous Referee #1 · 14 Feb 2017

In the manuscript "Effect of thermodenuding on the structure of nascent flame soot aggregates" authors present a study on nascent flame soot aggregates using thermodenuder (TD). The manuscript starts with introduction, where authors state the importance of atmospheric processing of soot particles for light scattering and absorption. Soot particle morphology is addressed to be one of the parameters that change optical particle properties. For this reason authors proposed a pathway for soot restructuring solely by thermodenuding due to weakening of the bonds between the aggregate monomers when no or minimal coating is present. However, the motivation of such study remained unclear. Authors did not provide any information on how abundant in atmosphere is nascent soot particles. Moreover, previous studies have already addressed soot restructuring due to surface tension effects. In those studies restructuring occurred solely when soot particles were coated with volatile materials,

which then were removed by thermodenuding. As it is very unlikely that soot particles in the atmosphere would remain unaltered by any aging processes, soot particles, if restructured, would change their morphology due to surface tension effects and not due to weakening of the bonds between the aggregate monomers. It suggests that this study is irrelevant for the atmospheric studies. Because of previous reasons, I cannot recommend this study for publishing in Atmospheric Measurement Techniques.

---

## Short Comment (SC1) · 15 Feb 2017

We thank the reviewer for the comments. We believe there was a misunderstanding on the goal and the scope of this study. We hope to clarify the misunderstanding with this reply. The purpose of the study was not to provide evidence (or lack of it) for a new pathway for atmospheric soot restructuring. The goal instead, was to study the potential bias introduced by a technique commonly used in atmospheric studies. In particular, we wanted to investigate the effect of thermodenuding on soot particles that, in purpose, had not been processed in the atmosphere yet (i.e., without any significant amount of coating). Thermodenuding is a technique used to study the effect that coating has on the optical properties of atmospheric soot. In fact, coating materials can have a substantial effect on the optical properties of soot particles; this aspect motivates our study – that is why we mentioned it in the introduction – but is not the

objective of the paper. Our sole purpose was to investigate whether the thermodenuding process itself changes the structure of soot or not. It is in fact, plausible that the thermodenuding process alone might restructure the refractory core of a mixed particle containing soot (a technique issue, not an atmospheric process). We agree with the reviewer that this process would not have a direct relevance to the atmosphere because this is an "artificial" manipulation of the particles. However, the process could be important for how data on thermodenuded particles, are interpreted. In other words, if thermodenuding is used to remove the coating material of atmospheric soot-containing particles, and if the denuding process artificially restructures the soot, then the measured effect would be a combination of the "natural" compaction due to the coating (either during condensation or evaporation) plus the "artificial" restructuring induced by the thermodenuder. This combination of factors would make the interpretation of atmospheric data complicated, at least. With this study, we wanted just to provide evidence that, fortunately, the thermodenuding process does not introduce this bias. This might seem as an insignificant result because it is, in a sense, a "null result". However, we feel it is a key information for the correct interpretation of ambient measurements. To conclude, we want to underline again, that this work does not address ambient processing at all but instead, it addresses the validity of a technique widely used in atmospheric studies. This is the reason why we submitted this manuscript to Atmospheric Measurement Techniques. We do believe this information can be very useful to those who routinely use a thermodenuder. However, because of the reviewer's comments, we feel that we might have failed to make this point completely clear in the manuscript. If given a chance, we will clarify the scope of the study in a revised version, after considering also the remaining reviewers' comments.

---

## Referee Comment (RC2) · Anonymous Referee #1 · 16 Feb 2017

Dear Authors,

For further discussion the following is needed:

1. A strong motivation, why pure (not covered in any substances) soot particles should get restructured after thermodenuding, is needed. It is not clear which processes should restructure soot agglomerates when it is heated with no other treatment. Moreover, soot vaporization temperature is approximately 4000K. Meaning that soot particles should remain stable at temperatures provided in this study.

2. Authors state: "In other words, if thermodenuding is used to remove the coating material of atmospheric soot-containing particles, and if the denuding process artificially restructures the soot, then the measured effect would be a combination of the "natural" compaction due to the coating (either during condensation or evaporation) plus

the "artificial" restructuring induced by the thermodenuder". I believe that the first part of this statement refers to previous studies on soot restructuring due to coating. The unclear part here is "...plus the "artificial" restructuring induced by the thermodenuder". In coating and denuding scenario, it is not clear why so called "artificial" restructuring should occur when the restructuring already occurred due to surface tension effects?

3. Authors state: "However, we feel it is a key information for the correct interpretation of ambient measurements". Can authors provide instances (references) when ambient measurements showed existence of pure, non-coated soot particles in the atmosphere? And when thermodenuding might have caused errors in estimating particle optical properties.

4. Studies on soot restructuring after particles were coated with other materials are plentiful. What are expected errors in light extinction due to restructuring?

---

## Author Comment (AC1) · 7 Apr 2017

We appreciate the insightful reviewers' comments. Below we address each comment separately and discuss changes in the new draft. Also changes made in the revised manuscript ( attached as a pdf file in supplement section below) are evidenced by using blue colored fonts. 1. A strong motivation, why pure (not covered in any substances) soot particles should get restructured after thermodenuding, is needed. It is not clear which processes should restructure soot agglomerates when it is heated with no other treatment. Moreover, soot vaporization temperature is approximately 4000K. Meaning that soot particles should remain stable at temperatures provided in this study. Thermal restructuring of soot at the thermodenuder temperature, is a hypothesis. If true, that would potentially bias the measured optical properties of soot when a thermodenuder is used on ambient particles. The main objective of this study is to test this hypothesis

to assure that the themodenuding process alone does not introduce this bias. Next we discuss some evidences of potential processes (not necessarily exclusive or independent) that could favor the compaction of lacy aggregates when thermodenuded. These evidences motivated us to study potential effects on the specific case of nascent soot: a) When heated, fractal-like aggregates of metal nanoparticles such, as silver, copper and metallic oxides (e.g., titania), have been found to restructure to more compact morphologies at temperatures well below the bulk material melting points. For example, thermal restructuring has been found in silver aggregates, even at temperatures as low as 100 oC, with full compaction at just 350 oC (much below the vaporization temperature of silver), while the primary particle size remained unchanged (Weber et al., 1996; Weber and Friedlander, 1997). Another study found that aggregates of titania started to collapse when temperatures reached 700 oC (Jang and Friedlander, 1998). These authors speculated that the heating causes the weakest branches in an aggregate to rotate around their contact points, resulting in the aggregate restructuring. Alternatively, Schmidt-Ott (1988) hypothesized that the monomers in silver nanoparticles aggregates might slide on each other when heated, causing compaction. Both processes would restructure the aggregates without complete breakage of the bonds between the monomers, which adhere to each other via Van der Waals forces.

b) Even in very controlled combustions, and depending upon the flaming conditions and fuel types, it is common for combustion generated soot aggregates to have different kinds of polycyclic aromatic hydrocarbons thinly coating them (Cross et al., 2010). In this case, coating acquired at the source, not added later through atmospheric processing. For this reason, soot freshly emitted is often referred to as "nascent" instead of "pure" soot. This coating on the nascent soot could play a role in determining the soot structure if the coating properties (i.e., viscosity and surface tension) change at the higher temperature of the thermodenuder. Chen et al. (2016) found that some polycyclic aromatic hydrocarbons like phenanthrene and flouranthene, when present as a submicron layer on soot, behaved as subcooled liquid that weakened the bonds between the monomers, allowing them to slide and roll over each other and resulting

in soot restructuring. Some evidence that thermodenuding might make a difference in the strength of the adhesive bonds between monomers is also provided by Rothenbacher et al. (Rothenbacher et al., 2008). For aged soot, they found that a higher degree of fragmentation was seen for thermodenuded particles (75% at 280 oC) than for untreated particles (60%) when impacted at $\sim$ 200 m/s. The degree of fragmentation was defined as the fraction of broken bonds in an aggregate. Although, the process involves both the effect of coating and impaction, the higher degree of fragmentation for thermodenuded particles suggests that the thermal energy has a role on the increased degree of fragmentation. We clarify these points in the introduction of the revised manuscript in 'page 4: line 6 through line 28'. 2. Authors state: "In other words, if thermodenuding is used to remove the coating material of atmospheric soot-containing particles, and if the denuding process artificially restructures the soot, then the measured effect would be a combination of the "natural" compaction due to the coating (either during condensation or evaporation) plus the "artificial" restructuring induced by the thermodenuder". I believe that the first part of this statement refers to previous studies on soot restructuring due to coating. The unclear part here is "...plus the "artificial" restructuring induced by the thermodenuder". In coating and denuding scenario, it is not clear why so called "artificial" restructuring should occur when the restructuring already occurred due to surface tension effects? Nascent soot particles can restructure during condensation or evaporation of coating material depending on the surface tension of the coating material (Ebert et al., 2002; Ma et al., 2013; Schnitzler et al., 2017; Tritscher et al., 2011). However, as we will discuss later on, the compaction might not be complete (e.g., not completely collapsed structure) and the thermal process, hypothetically (as from answer to comment 1.), could facilitate further restructuring. In fact, in the atmosphere there are several degrees of soot compaction. Some coating material indeed results in substantial, or even maximum compaction, but other coating materials actually result in negligible compaction. A clear example is shown in a laboratory study where particles coated with sulfuric acid did undergo severe restructuring, while soot particles coated by dioctyl sebacate showed only minimal or no

compaction (Cross et al., 2010). Evidences that soot compaction in the atmosphere comes in a continuum of values from low to high, are provided by several papers in the literature (e.g., China et al., 2015a; China et al., 2014; Radney et al., 2014; Zangmeister et al., 2014). Therefore, there could indeed be two sources of compaction, one due to coating naturally condensing on, or evaporating from, soot particles (which might not be full), as well as the hypothesized compaction due to the thermal energy provided during the thermodenuding process. As discussed in our response to comment 1, the main objective of this work was to find out whether or not, the thermal energy alone may facilitate the sliding/rolling of monomers at their point of contact, and restructure the soot aggregates, similarly to the cases of silver aggregates and titania aggregates discussed earlier on. Again, we felt that investigating this potential bias was important to assure the validity of a technique widely used. Fortunately, our result was negative, suggesting that the bias is negligible. We revise our manuscript expanding the discussion on soot restructuring. Firstly we report the compaction of soot by coating via condensation or evaporation of coating materials from previous studies on page 3 line 30 through line 34. Then, we put forward our hypothesis as "However, we hypothesize a third potential pathway for soot restructuring, in which restructuring might take place solely due to the thermodenuding process, through the added thermal energy. There are a few evidences that the thermodenuding process can favor the compaction of lacy aggregates, even in absence of coating material that condensed onto the primary emitted aggregates. If a similar process happens for ambient soot that would potentially bias the measured properties (e.g., absorption or scattering enhancements) of soot when a thermodenuder is used". Also we state the main objective our study as "The main objective of our study is to test this hypothesis to assure that the themodenuding process alone does not introduce this bias" and discuss a couple of potential restructuring processes based on previous results on other types of aggregates as in 'page 4: line 6 through line 28'.

3. Authors state: "However, we feel it is a key information for the correct interpretation of ambient measurements". Can authors provide instances (references) when

ambient measurements showed existence of pure, non-coated soot particles in the atmosphere? And when thermodenuding might have caused errors in estimating particle optical properties.

This point was partially addressed in response to comment 2. There are several studies that show that compaction of atmospheric soot particles comes in different degrees, likely linked to various atmospheric processes. We already provided some discussion on this topic in the original manuscript, but expanded the discussion here. As we mentioned in the response to comment 1, we do not think that realistic sources of soot, relevant to the atmosphere, would produce "pure" soot. So in our study we refer to "minimally coated soot", "nascent soot" or "freshly emitted soot". Several groups have shown that the degree of coating in atmospheric particles is very diverse (Adachi et al., 2010; China et al., 2013; China et al., 2014; China et al., 2015b; Healy et al., 2015; Liu et al., 2015; Zhang et al., 2008). These studies have been performed typically using electron microscopy, but also by other means, such as using mass and mobility measurements of atmospheric soot particles. Specifically, for the case of thinly coated soot, different studies show the presence of these particles in the atmosphere (China et al., 2015b; Wu et al., 2016). For example, in the study by China et al. (2014), a large fraction of freshly emitted soot particles collected on freeway on ramps were thinly coated (72%). In another field study, carried out at Pico Mountain Observatory in the Azores, China et al. (China et al., 2015b) found that 37% of the soot particles, in one sample, were thinly coated, even after days of atmospheric processing during the long-range transport in the free troposphere from the source. These two studies were carried out at very different locations (very near the source in the first study, and very far from the source in the second study) showing that thinly coated soot particles can exist in the atmosphere in very different environments. Based on the study of laboratory generated and ambient soot particles from two field campaigns: the 2010 CalNex study and Carbonaceous Aerosols and Radiative Effects Study (CARES), Cappa et al. (2012) observed only small increment in absorption enhancement (Eabs) for the ambient samples compared to the model values. Khalizov et al. (2013) suggested that

the discrepancy in measured value of (Eabs) from model may be due to the biased introduced by the use of themodenuder, when coating material is completely removed. They proposed that the thermodenuded soot particles can be more absorptive than the freshly emitted soot, which is always coated (at least thinly coated) and hence can introduce bias in getting the reference absorption for nascent soot.

We already provided some detail on this topic in the original manuscript based on existing literature, but we expanded this discussion in 'page 2: line 34 through 37' of the introduction of the revised manuscript.

4. Studies on soot restructuring after particles were coated with other materials are plentiful. What are expected errors in light extinction due to restructuring? We are aware of several papers (for example, Radney et al. (2014), China et al. (2015b), He et al. (2015), Wu et al. (2016)) that directly address the issue of changes in the optical properties due to soot compaction. A laboratory study was performed on soot compacted upon humidification; the study measured modest changes in the absorption cross section (5% to 14%) but the extinction cross section was much more sensitive to compaction, with changes in the order of more than 30% (Radney et al., 2014). Additionally, two previous studies from our group found similar results. These analyses were based on numerical simulations using the discrete dipole approximation on synthetic particles, closely emulating soot morphologies found in the atmosphere. Upon compaction, the simulation predicted a change in the absorption of a few percent, but a much more substantial change in the total scattering cross section (up to ∼300%) (China et al., 2015a; China et al., 2015b). Bond et al. (2009) and Mishchenko et al. (2004) recommend reducing the error in the aerosol single scattering albedo (the ratio of total scattering to the total extinction) to less than 3% (Bond et al., 2009; Mishchenko et al., 2004) for reducing our uncertainties on the direct effect of aerosols on climate. Therefore, we should attempt to minimize, or at least quantify, any potential source of bias. In addition to affecting the optical properties, changes in the soot compaction can also affect the results of laser induced incandescence measurements (Bambha et

al., 2013a; Bambha et al., 2013b). Condensation of secondary organic matter preferentially takes place in empty pores on soot particles and therefore, it is possible that compaction might affect secondary organic condensation on soot (Popovicheva et al., 2003). We further discuss the effect of soot compaction on scattering and absorption in 'page 3: line 21 through 30' of the introduction in the revised manuscript.

References Adachi, K., Chung, S. H., and Buseck, P. R.: Shapes of soot aerosol particles and implications for their effects on climate, Journal of Geophysical Research: Atmospheres, 115, 2010. Bambha, R., Dansson, M. A., Schrader, P. E., and Michelsen, H. A.: Effects of volatile coatings on the morphology and optical detection of combustion-generated black carbon particles, Sandia National Laboratories (SNL-CA), Livermore, CA (United States), 2013a. Bambha, R. P., Dansson, M. A., Schrader, P. E., and Michelsen, H. A.: Effects of volatile coatings and coating removal mechanisms on the morphology of graphitic soot, Carbon, 61, 80-96, 2013b. Bond, T. C., Covert, D. S., and Müller, T.: Truncation and angular-scattering corrections for absorbing aerosol in the TSI 3563 nephelometer, Aerosol Science and Technology, 43, 866-871, 2009. Cappa, C. D., Onasch, T. B., Massoli, P., Worsnop, D. R., Bates, T. S., Cross, E. S., Davidovits, P., Hakala, J., Hayden, K. L., and Jobson, B. T.: Radiative absorption enhancements due to the mixing state of atmospheric black carbon, Science, 337, 1078-1081, 2012. Chen, C., Fan, X., Shaltout, T., Qiu, C., Ma, Y., Goldman, A., and Khalizov, A. F.: An unexpected restructuring of combustion soot aggregates by subnanometer coatings of polycyclic aromatic hydrocarbons, Geophysical Research Letters, 43, 2016. China, S., Kulkarni, G., Scarnato, B. V., Sharma, N., Pekour, M., Shilling, J. E., Wilson, J., Zelenyuk, A., Chand, D., and Liu, S.: Morphology of diesel soot residuals from supercooled water droplets and ice crystals: implications for optical properties, Environmental Research Letters, 10, 114010, 2015a. China, S., Mazzoleni, C., Gorkowski, K., Aiken, A. C., and Dubey, M. K.: Morphology and mixing state of individual freshly emitted wildfire carbonaceous particles, Nature communications, 4, 2013. China, S., Salvadori, N., and Mazzoleni, C.: Effect of traffic and driving characteristics on morphology of atmospheric soot

particles at freeway on-ramps, Environmental science & technology, 48, 3128-3135, 2014. China, S., Scarnato, B., Owen, R. C., Zhang, B., Ampadu, M. T., Kumar, S., Dzepina, K., Dziobak, M. P., Fialho, P., and Perlinger, J. A.: Morphology and mixing state of aged soot particles at a remote marine free troposphere site: Implications for optical properties, Geophysical Research Letters, 42, 1243-1250, 2015b. Cross, E. S., Onasch, T. B., Ahern, A., Wrobel, W., Slowik, J. G., Olfert, J., Lack, D. A., Massoli, P., Cappa, C. D., and Schwarz, J. P.: Soot particle studies—instrument inter-comparison—project overview, Aerosol Science and Technology, 44, 592-611, 2010. Ebert, M., Inerle-Hof, M., and Weinbruch, S.: Environmental scanning electron microscopy as a new technique to determine the hygroscopic behaviour of individual aerosol particles, Atmospheric Environment, 36, 5909-5916, 2002. He, C., Liou, K.-N., Takano, Y., Zhang, R., Levy Zamora, M., Yang, P., Li, Q., and Leung, L. R.: Variation of the radiative properties during black carbon aging: theoretical and experimental intercomparison, Atmospheric Chemistry and Physics, 15, 11967-11980, 2015. Healy, R. M., Wang, J. M., Jeong, C. H., Lee, A. K., Willis, M. D., Jaroudi, E., Zimmerman, N., Hilker, N., Murphy, M., and Eckhardt, S.: Light‐absorbing properties of ambient black carbon and brown carbon from fossil fuel and biomass burning sources, Journal of Geophysical Research: Atmospheres, 120, 6619-6633, 2015. Jang, H. D. and Friedlander, S. K.: Restructuring of chain aggregates of titania nanoparticles in the gas phase, Aerosol Science and Technology, 29, 81-91, 1998. Khalizov, A. F., Lin, Y., Qiu, C., Guo, S., Collins, D., and Zhang, R.: Role of OH-initiated oxidation of isoprene in aging of combustion soot, Environmental science & technology, 47, 2254-2263, 2013. Liu, S., Aiken, A. C., Gorkowski, K., Dubey, M. K., Cappa, C. D., Williams, L. R., Herndon, S. C., Massoli, P., Fortner, E. C., and Chhabra, P. S.: Enhanced light absorption by mixed source black and brown carbon particles in UK winter, Nature communications, 6, 2015. Ma, X., Zangmeister, C. D., Gigault, J., Mulholland, G. W., and Zachariah, M. R.: Soot aggregate restructuring during water processing, Journal of Aerosol Science, 66, 209-219, 2013.

[Figure]

Please also note the supplement to this comment:
http://www.atmos-meas-tech-discuss.net/amt-2016-270/amt-2016-270-AC1-supplement.pdf

**Supplement:**

**Effect of thermodenuding on the structure of nascent flame soot aggregates**

[revised manuscript text omitted]

We finally discuss the assumption 4), which is the focus of our study. Previous studies have shown that nascent soot particles can restructure during the condensation or evaporation of the coating material depending on their surface tension (e.g., Ebert et al., 2002; Ma et al., 2013; Tritscher et al., 2011). Xue et al. (2009) found significant restructuring of soot particles when the particles were first coated with glutaric acid and then denuded. Ghazi and Olfert (2013) reported the dependence of soot restructuring on the mass amount of different coating material types. This restructure alone can affect the soot particles optical properties. For example, a laboratory study was performed on soot compacted upon humidification; the study measured modest changes in the absorption cross section (5% to 14%) but the extinction cross section was much more sensitive to compaction, with changes of more than 30% (Radney et al., 2014). Similarly, China et al. (2015a) and China et al. (2015b) using numerical simulations, predicted small changes in the absorption cross section (a few percent), but a much more substantial change in the total scattering cross section (up to ~300%), upon soot compaction. In addition to affecting the optical properties, changes in the soot structure can also affect the results of laser induced incandescence measurements (Bambha et al., 2013a). Finally, the condensation of secondary organic matter preferentially takes place in empty pores on soot particles and therefore, it is possible that compaction might affect secondary organic condensation on soot (Popovicheva et al., 2003). Two potential explanations for the soot restructuring detected during these studies can be: 1) Soot might be compacted during condensation of the coating materials due to surface tension effects (Huang et al., 1994; Schnitzler et al., 2017; Tritscher et al., 2011; Zhang et al., 2008). 2) The removal of the coating material during the subsequent thermodenuding may cause compaction when the coating evaporates, still due to surface tension effects (Ebert et al., 2002; Ma et al., 2013). However, we hypothesize a third potential pathway for soot restructuring, in which restructuring might take place solely due to the thermodenuding process, through the added thermal energy. There are a few evidences that the thermodenuding process can favor the compaction of lacy aggregates, even in absence of

coating material that condensed onto the primary emitted aggregates. If a similar process happens for ambient soot that would potentially bias the measured properties (e.g., absorption or scattering enhancements) of soot when a thermodenuder is used. The main objective of our study is to test this hypothesis to assure that the themodenuding process alone does not introduce this bias. A couple of potential restructuring processes, induced while thermodenuding aggregates, are discussed next:

a)        When heated, fractal-like aggregates of metal nanoparticles, such as silver, copper and metallic oxides (e.g., titania), have been found to restructure to more compact morphologies at temperatures well below the bulk material melting points. For example, thermal restructuring has been found in silver aggregates, even at temperatures as low as 100 ºC, with full compaction at just 350 ºC (much below the melting temperature of silver),  while the primary particle size remained unchanged (Weber et al., 1996; Weber and Friedlander, 1997). Another study found that aggregates of titania started to collapse when temperatures reached 700 ºC (Jang and Friedlander, 1998). These authors speculated that the heating causes the weakest branches in an aggregate to rotate around their contact points, resulting in the aggregate restructuring. Alternatively, Schmidt-Ott (1988) hypothesized that the monomers in silver nanoparticles aggregates might slide on each other when heated, also causing compaction. Both processes would restructure the aggregates without complete breakage of the bonds between the monomers due to Van der Waals forces.

b)        As mentioned earlier, nascent soot aggregates typically have polycyclic aromatic hydrocarbons thinly coating them. This nascent coating could play a role in determining the soot structure if the coating properties (i.e., viscosity and surface tension) change at the higher temperature of the thermodenuder. Chen et al. (2016) found that some polycyclic aromatic hydrocarbons like phenanthrene and flouranthene, when present as a subnanometer layer on soot, behaved as subcooled liquid that weakened the bonds between the monomers, allowing them to slide and roll over each other and resulting in soot restructuring. Rothenbacher et al. (2008) provided some evidences that thermodenuding might make a difference in the strength of the adhesive bonds between the monomers. For aged soot, they found that a higher degree of fragmentation was seen for thermodenuded particles (75% at 280 ºC) than for untreated particles (60%) when impacted at ~ 200 m/s. The degree of fragmentation was defined as the fraction of broken bonds in an aggregate. Although, the process involved both the effect of coating and impaction, the higher degree of fragmentation for thermodenuded particles suggests that the thermal energy has a role on the increased degree of fragmentation.

These evidences motivated us to study potential effects on the specific case of nascent soot. With this goal in mind, we analyze 
[revised manuscript text omitted]

**4 Summary**

In this study, we used scanning electron microscopy to investigate the morphology of nascent soot aggregates prior and after thermodenuding in a low-temperature regime (< 270 ºC). Despite mass losses up to ~29% in the nascent soot, only minor effects on the soot structure were detected after thermodenuding, irrespective of the fuel type. No significant change in $D_f$ was observed; the only exception was the fractal dimension of nascent-oxidized soot, although roundness and convexity showed only minor changes also in this case. Future work should focus on the effect of higher thermodenuding temperatures to investigate temperature effects on the structure of nascent soot.

*The image analysis data used in this paper are publicly available on the Digital Commons repository of Michigan Technological University and can be found here: http://digitalcommons.mtu.edu/physics-fp/80*

**Acknowledgements**

This work was supported in part by the Office of Science (BER), Department of Energy (Atmospheric System Research) Grant no. DE-SC0011935 and no. DE-SC0010019, and Atmospheric Chemistry program of the National Science Foundation Grant no. AGS-1536939 to Boston College, 1537446 to Aerodyne Research Inc. S. China was
5   partially supported by a NASA Earth and Space Science Graduate Fellowships no. NNX12AN97H.

[revised manuscript text omitted]